# pH and Reduction Dual-Responsive Bi-Drugs Conjugated Dextran Assemblies for Combination Chemotherapy and In Vitro Evaluation

**DOI:** 10.3390/polym13091515

**Published:** 2021-05-08

**Authors:** Xiukun Xue, Yanjuan Wu, Xiao Xu, Ben Xu, Zhaowei Chen, Tianduo Li

**Affiliations:** 1Shandong Provincial Key Laboratory of Molecular Engineering, School of Chemistry and Chemical Engineering, Qilu University of Technology (Shandong Academy of Science), Jinan 250353, China; Jasonxue@yeah.net (X.X.); xuben2019@126.com (B.X.); 2Institute of Food Safety and Environment Monitoring, College of Chemistry, Fuzhou University, Fuzhou 350108, China; 201310023@fzu.edu.cn

**Keywords:** polymeric prodrug, dual-sensitive, combination chemotherapy, drug conjugation, dextran

## Abstract

Polymeric prodrugs, synthesized by conjugating chemotherapeutic agents to functional polymers, have been extensively investigated and employed for safer and more efficacious cancer therapy. By rational design, a pH and reduction dual-sensitive dextran-di-drugs conjugate (oDex-g-Pt+DOX) was synthesized by the covalent conjugation of Pt (IV) prodrug and doxorubicin (DOX) to an oxidized dextran (oDex). Pt (IV) prodrug and DOX were linked by the versatile efficient esterification reactions and Schiff base reaction, respectively. oDex-g-Pt+DOX could self-assemble into nanoparticles with an average diameter at around 180 nm. The acidic and reductive (GSH) environment induced degradation and drug release behavior of the resulting nanoparticles (oDex-g-Pt+DOX NPs) were systematically investigated by optical experiment, DLS analysis, TEM measurement, and in vitro drugs release experiment. Effective cellular uptake of the oDex-g-Pt+DOX NPs was identified by the human cervical carcinoma HeLa cells via confocal laser scanning microscopy. Furthermore, oDex-g-Pt+DOX NPs displayed a comparable antiproliferative activity than the simple combination of free cisplatin and DOX (Cis+DOX) as the extension of time. More importantly, oDex-g-Pt+DOX NPs exhibited remarkable reversal ability of tumor resistance compared to the cisplatin in cisplatin-resistant lung carcinoma A549 cells. Take advantage of the acidic and reductive microenvironment of tumors, this smart polymer-dual-drugs conjugate could serve as a promising and effective nanomedicine for combination chemotherapy.

## 1. Introduction

Nanosized drug delivery systems (nDDSs) with an ability to load multiple chemotherapeutic agents simultaneously have gained considerable interest for precise and efficient cancer therapy [1,2,3,4,5,6]. The nDDSs currently developed for combination chemotherapy include inorganic nanoparticles (NPs), liposomes, organic-inorganic hybrid materials, and polymeric NPS, etc. [7,8,9]. Among various nDDSs, polymeric drug delivery systems have played an integral role due to several additional advantageous properties, such as biocompatibility/biodegradability, synthetic flexibility, and tailorable properties [10,11,12,13]. Encapsulation of dual drugs into polymeric NPs or conjugation of one drug onto the polymer backbones while loading another drug into the formed nanosystems are among the most pervasive strategies [14,15,16,17,18]. As an example, recently, a novel dual-drugs delivery system composed of nanocrystalline cellulose and L-lysine were developed for efficient co-encapsulation of model chemotherapeutic curcumin and methotrexate [19]. In another study, disulfide-containing camptothecin (CPT) was conjugated to poly (L-glutamic acid)-graft-methoxy poly (ethylene glycol) (PLG-g-mPEG) via esterification reaction for the preparation of an amphiphilic biodegradable prodrug (PLG-g-mPEG/CPT) [18]. The prodrug could co-assemble with DOX into micellar NPs for combination cancer therapy. Previous reports demonstrated the application of these polymeric nDDSs for combination chemotherapy could simply conquer the problem of poor water solubility of most hydrophobic chemotherapeutic agents, improve their biodistribution, enhance treatment efficiency, and reduce the systemic toxicity. Moreover, an effective way to further enhance the chemotherapeutic efficiency is to adopt tumor site-specific responsive linkage for drug conjugation. Specifically, it is well-known that the corresponding pH value of tumor extracellular microenvironment, endosomes, and lysosomes is approximately 6.8, 5.5–6.0, and 4.5–5.0, which is lower than the physiological pH 7.4 [20,21,22]. Thus, various acidic pH-cleavable linkages, such as hydrazone, orthoester, acetal, and carbamate have been successfully designed and synthesized for drug conjugation to realize efficient intracellular drug release [23]. In addition, the high concentration gradient of glutathione (GSH), a thiol-containing tripeptide, between the intracellular (1–10 mM) and extracellular areas (2–20 μM) is frequently used as a reductive stimuli to boost the drug release from nanocarriers [24]. A series of reduction-responsive linkages have been developed for drug conjugation. For example, CPT was conjugated to the side chains of poly (methacrylate) via thioether bond, which could be cleaved by GSH via thiolysis [24]. Notably, a prominent hallmark of tumor cell is the heterogeneous coexistence of lower pH environment and overproduced intracellular GSH compared with those in blood and normal cells [24,25]. Accordingly, the pH and reduction dual-sensitive prodrug nDDSs have recently received increasing attention.

Dextran, with α-1,6-glycosidically linked glucose chains, is a natural and highly water-soluble polymer, which has great utility in biomedical applications. Dextran has many hydroxyl groups that can be readily chemically modified. Importantly, dextran is a kind of FDA-approved natural polymer. Several studies have reported the use of dextran and its derivatives as nanocarriers for cancer imaging and therapy [26,27,28,29,30]. Herein, we prepare a novel pH and reduction dual-sensitive polymeric prodrug oDex-g-Pt+DOX, in which Pt (IV) prodrug was conjugated to the oxidized dextran (oDex) as reduction-sensitive segment, and then DOX was conjugated via acid-cleavable hydrazone bond (Scheme 1). The amphiphilic oDex-g-Pt+DOX could further self-assemble into well-defined NPs in aqueous solution with the average diameters of ∼180 nm. Considering that DOX can restrain the DNA remodeling, and further inhibit the repairment of cisplatin-damaged duplex DNA by suppressing the activity of topoisomerase II, simultaneous delivery of Pt (IV) and DOX may work in a synergistic way to overcome drug resistance. Our results revealed that this novel polymeric prodrug oDex-g-Pt+DOX NPs presented pH and reduction dual-triggered drug release behavior, could effectively kill cancer cells after intracellular internalization, and reverse cisplatin resistance in cisplatin-resistant lung carcinoma A549 cells (A549/DDP cells). 

## 2. Materials and Methods 

### 2.1. Materials

Dextran10 k, 3-(4,5-dimethylthiazol-2-yl)-2,5-diphenyltetrazolium bromide (MTT, 98%), sodium metaperiodate (NaIO_4_, ≥99%), doxorubicin hydrochloride (DOX·HCl, 98%), and Hoechst 33,258 (≥98%) was obtained from Sigma-Aldrich (Shanghai, China). Cisplatin (99.8%) was bought from Shandong Boyuan Pharmaceuti-cal Co., Ltd. (Jinan, Shandong, China). Hydrogen peroxide solution (H_2_O_2_, 30 wt% in water), succinic anhydride (99%), triethylamine (TEA, ≥99.5%), and 1-ethyl-3-(3-dimethylaminopropyl) carbodiimide hydrochloride (EDC·HCl, ≥98%) were purchased from Aladdin Chemistry Co. Ltd. (Shanghai, China). Phosphate-buffered saline (PBS), dulbecco’s modified eagle’s medium (DMEM)/high-glucose medium, and trypsin were obtained from GE Healthcare Life Sciences (Beijing, China). Gibco Fetal Bovine Serum (FBS) was purchased from Thermo Fisher Scientific Inc. (Shanghai, China). Lyso-tracker was purchased from Shanghai Biyuntian Biological Co., Ltd. (Shanghai, China). All chemical reagents were used as received without purification if not mentioned otherwise. Dimethyl sulfoxide (DMSO, ≥99%) was purchased from Sinopharm Chemical Reagent Co., Ltd. (Shanghai, China), and distilled followed by dried with calcium hydride (CaH_2_) for 2 weeks.

### 2.2. Measurements

Proton nuclear magnetic resonance (^1^H NMR) spectra were collected at room temperature (rt) on a Bruker AVANCE II 400 NMR spectrometer using deuterated dimethyl sulfoxide (DMSO-d_6_) as solvent. Fourier transform infrared (FT-IR) spectra were measured on a Thermo Scientific Nicolet IS10 instrument. The platinum content was recorded on an inductively coupled plasma mass spectrometer (ICP-MS) and inductively coupled plasma optical emission spectrometer (ICP-OES, Thermoscientific, USA). UV-visible absorption spectra were used for quantitative determination of the DOX concentration by a Cary 5000 UV-Vis-NIR spectrometer. The diameters and polydispersity index (PDI) measurements were performed at rt on a Malvern Zetasizer Nano ZS90 instrument with a vertically polarized He-Ne laser. The morphology of the NPs was investigated by transmission electron microscopy (TEM) using a JEOL JEM 2100 electron microscope. Confocal laser scanning microscopy (CLSM) micrographs were visualized with a Leica SP8 CLSM image system. Quantitative DOX fluorescent analysis of cellular uptake was investigated by BD FACSCalibur flow cytometry imaging system.

### 2.3. Synthesis of Oxidized Dextran (oDex)

Firstly, Dex (2.0 g) was dissolved in 20 mL distilled water under 50 °C for 20 min. Then, NaIO_4_ solution (53 mg·mL^−1^, 20 mL) was added to the above transparent solution for oxidation process. After stirring for 6 h at rt, 2 mL ethylene glycol was added in order to stop the oxidation reaction. The mixture was purified by dialysis (MWCO: 7000 Da) against distilled water for 3 days and finally lyophilized.

### 2.4. General Procedure for Synthesis of Platinum Conjugated oDex (oDex-g-Pt)

The synthesis of diamminedichloro-dihydroxyplatinum (c,c,t-[Pt(NH_3_)_2_Cl_2_(OH)_2_], DHP) and succinic anhydride modified DHP (c,c,t-[Pt(NH_3_)_2_Cl_2_(OH)(O_2_CCH_2_CH_2_CH_2_CO_2_H)], Pt (IV) prodrug) were presented in the Appendix A according to the previous reports (Appendix A) [31]. In general, oDex (1.0 g), and Pt (IV) (1085 mg, 2.5 mmol) were dissolved in dried DMSO (40 mL) under argon, then EDC·HCl (4793 mg, 25 mmol) and DMAP (305 mg, 2.5 mmol) were added. The mixture was vigorously stirred at rt for 72 h while the argon inert atmosphere was maintained. The solution was dialyzed against distilled water for 48 h using a pre-swelled Spectra/Por Regenerated Cellulose membrane (MWCO: 3500 Da) to remove DMSO, unreacted Pt (IV), and condensation reagents. Finally, the light yellow solution was freeze-dried to obtain oDex-g-Pt conjugates.

### 2.5. Conjugation of DOX with oDex-g-Pt

DOX was incorporated onto the pendants of oDex-g-Pt through a hydrazone bond. DOX·HCl (10 mg) was dissolved in dried DMSO (1 mL) with the help of TEA (20 µL) to remove HCl. The oDex-g-Pt (100 mg) was also dissolved in 10 mL of dried DMSO. Then, those two solutions were mixed and reacted at rt in dark for another 72 h. Afterwards, the mixture was dialyzed (MWCO: 3500 Da) against DMSO for 24 h to remove unconjugated DOX, and then against the phosphate buffer saline (PBS, pH 7.4) for 48 h to remove DMSO. Finally, the DOX conjugated oDex-g-Pt product (oDex-g-Pt+DOX) was obtained as dark red powder after lyophilization. oDex-g-DOX, a control, was similarly prepared by covalent conjugation of DOX to the oDex (Appendix A).

### 2.6. Preparation and Characterization of oDex-g-Pt and oDex-g-Pt+DOX NPs

Briefly, oDex-g-Pt+DOX (10 mg) was dissolved in DMSO (2 mL), and stirred at rt in dark for 12 h. The solution was dropwise injected into Milli-Q water (8 mL) within 10 min and continuously stirred for another 4 h. Then the suspension was dialyzed (MWCO: 3500 Da) against Milli-Q water for 48 h to prepare the oDex-g-Pt+DOX NPs. The oDex-g-Pt NPs and oDex-g-DOX NPs were also prepared similarly. The morphology, diameters and PDI of oDex-g-DOX NPs, oDex-g-Pt NPs and oDex-g-Pt+DOX NPs in Milli-Q water were determined by TEM and DLS measurements. The content of Pt in the oDex-g-Pt NPs and oDex-g-Pt+DOX NPs were measured by ICP-OES. UV−vis spectra of DOX and oDex-g-Pt+DOX NPs were recorded. The content of DOX in the oDex-g-Pt+DOX NPs was then calculated by using a DOX standard curve. The Pt and DOX loading content (DLC) were calculated as following equations:DLC (%) = [drug in NPs/total weight of NPs] × 100%(1)

### 2.7. pH- and Reduction-Sensitivity of oDex-g-Pt+DOX NPs

Changes in the particle sizes and size distributions of oDex-g-Pt+DOX NPs were observed using DLS and TEM measurements after incubation in PBS (pH = 7.4, 0.1 M) buffer containing GSH at 0 mM and 10 mM. The colloidal stability of oDex-g-Pt+DOX NPs in acetate buffer solution (ABS, pH = 5.0) containing GSH at 0 mM and 10 mM was also characterized. In a typical experiment, oDex-g-Pt+DOX NPs (1 mg/mL) were prepared in PBS buffer (pH = 7.4) as above, GSH was added at specified concentrations with continuous stirring. At predetermined time intervals, 3 mL of sample was used for DLS characterization, the average diameter and PDI were recorded. Moreover, the morphology changes of oDex-g-Pt+DOX NPs were analyzed by TEM. Tyndall effect of oDex-g-Pt+DOX NPs in pH 7.4 and pH 5.0 with 10mM GSH for 24 h were also measured.

### 2.8. pH- and Reduction-Activated Drugs Release from oDex-g-Pt+DOX NPs

The DOX and Pt release profiles from oDex-g-Pt+DOX NPs were investigated using a dialysis technique. In general, the lyophilized oDex-g-Pt+DOX NPs (2 mg) was resuspended in 2 mL of PBS (0.1 M, pH = 7.4), PBS (0.1 M, pH = 7.4) containing 10 mM GSH, ABS (0.1 M, pH = 5.0), or ABS (0.1 M, pH = 5.0) containing 10 mM GSH. Samples were placed into Spectra/Por Regenerated Cellulose membrane (MWCO: 3500 Da), and immersed into the corresponding buffer solutions (18 mL). Dialysis was continued in a shaking culture incubator at 37 °C, and 2 mL aliquots were sampled and replenished with equivalent volumes of fresh buffer at desired time points. Concentrations of DOX released into the aqueous solution from oDex-g-Pt+DOX NPs in acidic and reductive environments were quantitatively measured using UV−vis spectroscopy, and the percentages of DOX release were calculated by interpolation from its standard calibration curve. Simultaneously, the Pt concentration was detected via ICP-MS after nitrolysis. The drugs released from oDex-g-Pt+DOX NPs were expressed as percentage of cumulative drugs in the solution to the total drugs in the oDex-g-Pt+DOX NPs as a function of time.

### 2.9. Cell Culture

Human cervical carcinoma HeLa cells, normal fibroblasts L929 cells, lung carcinoma A549 cells, and A549/DDP cells were obtained from the institute of biochemistry and cell biology, Chinese academy of sciences (Shanghai, China). The cell culture medium was DMEM containing 10% FBS and 100 IU/mL penicillin-streptomycin, and was replaced every 2 days to keep the exponential growth of cells. The cells were incubated under 5% CO_2_ at 37 °C.

### 2.10. In Vitro Cytotoxicity

The MTT assay was used to assess the biocompatibility of dextran and oDex against L929 and HeLa cells according to the previous reports [10]. For L929 test, cells at 5 × 10^3^ cells/well were placed into 96-plates and incubated with DMEM (100 μL) for 24 h. Then, the culture medium was replenished with 200 μL of fresh DMEM medium containing different amounts of dextran or oDex (31.25, 62.5, 125, 250, 500 and 1000 μg/mL) for another 48 h incubation, respectively. MTT solution (20 μL, 5 mg/mL in PBS) was added to each well for another 4 h co-incubation at 37 °C, followed by removal of solution containing MTT. DMSO (150 μL) was added to each well to dissolve the purple formazan crystals formed by live cells. Finally, after shaking for 10 min, the absorbance of purple formazan product was recorded at 490 nm by a microplate reader. The cellular viability was calculated according to the following equation:Cellular viability (%) = [OD of the treated cells/OD of control cells] × 100%(2)(optical density: OD).

The nano formulations oDex-g-DOX NPs, oDex-g-Pt NPs, oDex-g-Pt+DOX NPs, free cisplatin (Cis), free DOX, Pt (IV) and the combinational forms were also evaluated for in vitro cytotoxicity utilizing HeLa, A549, and A549/DDP cell lines by the standard MTT assay. Briefly, after HeLa cells were cultivated as above, the culture medium was replaced by Cis, DOX, Pt (IV), Cis + DOX, Pt (IV) + DOX, oDex-g-Pt NPs, and oDex-g-Pt+DOX NPs at a final DOX concentration from 2.437–156 μM or Pt concentration from 3.375−216 μM for 48 h or 72 h of incubation. Then, MTT assay was completed as above.

The half maximal inhibitory concentration (IC_50_ value) is the drug concentration that can inhibit or kill 50% of the cells. 

### 2.11. In Vitro Cellular Uptake and Intracellular Distribution

The in vitro cellular uptake efficiency and intracellular drugs distribution of oDex-g-Pt+DOX NPs was investigated in HeLa cells using the CLSM technique. HeLa cells were seeded on the microscope slides in a 6-well plate with 3 × 10^5^ cells/well with DMEM medium (2 mL), and cultured overnight at 37 °C in a humidified atmosphere containing 5% CO_2_. Then, the medium was removed, 2 mL of oDex-g-Pt+DOX NPs solution (5 μg DOX/mL) was added. After further 0.5 h or 4 h incubation, cells were gently rinsed with cold PBS. Subsequently, cells were co-incubated with Lyso-tracker (1 mL, 50 nM) medium for another 30 min. After that, the cells were washed twice with cold PBS (10 mM, pH = 7.4) and fixed with paraformaldehyde solution (4% in 10 mM PBS, pH = 7.4) for 20 min. Then, the cell nuclei were dyed with Hoechst 33,258 (10 μg/mL, 1 mL) for 10 min, and the in vitro intracellular fluorescence was recorded for Hoechst 33,258 (405 nm), DOX (480 nm), and Lyso-tracker (555 nm) on a Leica SP8 CLSM image system.

Flow cytometric analysis was also used to evaluate the in vitro cellular uptake efficiency of free DOX and oDex-g-Pt+DOX NPs. For flow cytometry measurements (FCM), HeLa cells were planted into 12-well plates at a density of 1 × 10^5^ cells/well and cultured for 24 h. Subsequently, the DMEM culture medium was discarded, and cells were treated with free DOX or oDex-g-Pt+DOX NPs a fixed DOX-equivalent concentration, followed by 0.5 h or 4 h incubation. Afterwards, the cells were washed with PBS (10 mM, pH = 7.4) and detached by trypsin. After centrifugation, the cells were resuspended in cold PBS (10 mM, pH = 7.4, 0.5 mL) for flow cytometry measurements.

### 2.12. Statistical Analysis

All the experiments had three replicates (*n* = 3) at least. Data were presented as mean standard deviation (SD). The one-way ANOVA analysis with a Tukey post-hoc test was carried out to analyze the statistical significance: * *p* < 0.5, ** *p* < 0.01, *** *p* < 0.001, **** *p* < 0.0001.

## 3. Results

### 3.1. Design, Synthesis and Characterization of Drug, Drug-Polymer Conjugates

Dextran, a kind of hydrophilic and biocompatible polymer with massive side hydroxyl groups, has been extensively studied for biomedical applications, such as antibacterial biofilm, drug delivery particles, bioink for 3D printing, and so on [21,32]. DOX is one of the most commonly used and effective antineoplastic drugs with outstanding activity against various solid tumors. DOX can inhibit the synthesis of DNA by intercalating into DNA duplex and further inhibiting nucleic acid biosynthesis. Cisplatin, a leading platinum-based chemotherapeutic drug, has been widely applied for the treatment of various cancers, including breast, ovarian, head and neck, bladder, thyroid, prostate, and non small cell lung cancer [33]. Although cisplatin has gained great success in clinical cancer treatment, there are still some crucial concerns associated with the use of cisplatin, on account of cisplatin resistance and significant side effects. Many cancer cells have intrinsic and/or quickly acquired resistance, leading to unsatisfactory therapeutic efficacy. Meanwhile, cisplatin has severe side effects to the normal tissues, particularly chronic neurotoxicity and acute nephrotoxicity, which is mainly due to the lack of specificity. More importantly, previous reports have indicated that the combination of DOX and cisplatin could enhance the anticancer efficiency, decrease the side effects, and further reverse the cisplatin resistance [34,35]. As shown in Scheme 2, in our design, dextran was oxidized by NaIO_4_ to prepare the oDex with abundant of side hydroxyl and aldehyde groups. Subsequently, Pt (IV) prodrug, a cisplatin derivative, and DOX were gradually grafted to oDex through coupling reactions to prepare oDex-g-Pt+DOX conjugates. The synthesized molecules and polymers were listed in Table 1.

Initially, DHP was functionalized with succinic anhydride to synthesize a Pt (IV) prodrug with one carboxyl group as axial ligand. The structures of DHP, Pt (IV) prodrug, and oDex-g-DOX were characterized using ^1^H NMR and FT-IR spectra (Appendix A). The chemical structures of oDex, oDex-g-Pt, and oDex-g-Pt+DOX were also confirmed by both ^1^H NMR (Figure 1) and FT-IR spectra (Appendix A). As shown in Figure 1, ^1^H NMR spectrum of oDex exhibited a new peak at δ 9.6 ppm, which was the characteristic resonance signal of the aldehyde group (-CHO). The successful synthesis of oDex was further confirmed by FT-IR spectrum, a new absorption band at 1734 cm^−1^ assigned to the aldehyde group in oDex (Appendix A). In oDex-g-Pt, a new proton resonance peak that arose at δ 2.4 ppm was be ascribed to methylene group adjacent to Pt, which confirmed the graft of succinic anhydride modified DHP (Pt (IV) prodrug) to oDex. For oDex-g-Pt+DOX, the peaks at δ 7.0–8.0 ppm were the signals of benzene protons from DOX segment, and the new stretching vibration located at 1635 cm^−1^ (ν−C=N−) was attributed to imine bond, demonstrated the successful synthesis. Moreover, the DLC of DOX and Pt in oDex-g-Pt+DOX were calculated to be 11.26% and 5.89%, respectively. The precise chemical structures of drugs and drugs-oDex conjugates ensure the possibility of the clinical applications. Furthermore, the UV-Vis spectra of free DOX and oDex-g-Pt+DOX were also measured. As shown in Appendix A, the absorption peak of DOX could be clearly observed in oDex-g-DOX and oDex-g-Pt+DOX system, which indicated the existence of DOX.

### 3.2. Preparation and Characterization of oDex-g-Pt NPs and oDex-g-Pt+DOX NPs

In our work, Pt (IV) and DOX acted not only as chemotherapeutic drugs, but also as hydrophobic segment in oDex-g-Pt, oDex-g-DOX and oDex-g-Pt+DOX. Meanwhile, oDex had a great quantity of hydroxyl groups, which were hydrophilic. It is reasonable that oDex-g-Pt, oDex-g-DOX and oDex-g-Pt+DOX could self-assemble into NPs in aqueous medium. oDex-g-Pt NPs, oDex-g-DOX and oDex-g-Pt+DOX NPs were prepared using a simple dialysis method. Particle size and size distribution play significant roles in intravenous activities. Figure 2A,B showed the DLS results of oDex-g-Pt NPs and oDex-g-Pt+DOX NPs. oDex-g-Pt NPs had an average particle size of 95 nm and a low PDI of 0.17 in Milli-Q water. Under same condition, oDex-g-Pt+DOX NPs had a mean particle size of 183 nm and a PDI of 0.22. The size and size distribution of control group, oDex-g-DOX, were displayed in Appendix A. In addition, TEM micrographs indicated that both oDex-g-Pt NPs and oDex-g-Pt+DOX NPs had clear spherical morphology (Figure 2C,D). Notably, the size of oDex-g-Pt NPs and oDex-g-Pt+DOX NPs determined by TEM was smaller than DLS results due to the swelling of NPs in aqueous medium for DLS measurements. All the results demonstrated that both oDex-g-Pt and oDex-g-Pt+DOX could successfully assemble into small and uniform NPs. The appropriate sizes of oDex-g-Pt+DOX NPs benefited the increased intratumoral accumulation via the enhanced permeability and retention effect [26]. Moreover, the zeta potential values of oDex-g-Pt and oDex-g-Pt+DOX were −3.18 mV and 5.80 mV, respectively (Appendix A), which also demonstrated the successful conjugation of DOX.

### 3.3. Stability of oDex-g-Pt+DOX NPs

For oDex-g-Pt+DOX NPs, Pt (IV) prodrug was conjugated to the oDex via the esterification reaction, and DOX was introduced via the schiff base reaction. As is well known, Pt (IV) prodrug can be simply reduced into active Pt (II) drugs by reductive agents, such as sodium ascorbic, DTT and GSH, and the imine linkage is sensitive to the acidic environment. Consequently, the oDex-g-Pt+DOX NPs can respond well to the tumor acidic and reductive microenvironment, and then quickly dissociate to release active DOX and Pt (II) drugs. 

Initially, the average size and size distribution changes of oDex-g-Pt+DOX NPs in response to pH and GSH were followed by DLS measurements. DOX was conjugated onto the oDex-g-Pt via Schiff base reaction, the formed hydrazone linkages are acidity-labile. Then, two pH values, pH 5.0 and pH 7.4, were selected for the stability evaluation of the prepared oDex-g-Pt+DOX NPs. pH 5.0 and pH 7.4 were used to mimic the tumor intracellular components and normal physiological conditions, respectively. In Figure 3A,B, no significant variations in mean size and PDI were observed for oDex-g-Pt+DOX NPs in PBS at physiological pH (pH = 7.4) over 48 h, which indicated that oDex-g-Pt+DOX NPs had high stability under such conditions. However, oDex-g-Pt+DOX NPs underwent gradual swelling at pH 5.0, in which the average diameter increased from ca. 204 to 372 nm, and the PDI value was in a constant state of change. These changes might be ascribed to the acid-triggered release of DOX, which induced the instability of the NPs. To evaluate reduction-sensitivity of oDex-g-Pt+DOX NPs, the NPs were incubated in PBS (pH = 7.4) with 10 mM GSH. The results showed that the size and size distribution were constantly changing with increase in incubation time. Generally, both the average diameter and PDI increased. These changes suggested the initial reduction of the Pt (IV) groups increasing the hydrophilicity of oDex-g-Pt+DOX NPs, and consequent disassembly and re-assembly of the NPs. Notably, DOX was still conjugated to oDex as hydrophobic segment. Certainly, the disassembly and re-assembly could also be detected in PBS (pH = 5.0) with 10 mM GSH. As shown in Figure 3A,B, the average size increased from ca. 251 to 703 nm within 15 h, and then progressively decreased to 289 nm in 48 h. Simultaneously, the PDI value showed a similar phenomenon. In particular, size and size distribution of the oDex-g-Pt+DOX NPs incubated in PBS (pH = 5.0), PBS (pH = 7.4) with 10 mM GSH, and PBS (pH = 5.0) with 10 mM GSH for 24 h are shown in Figure 3C–E, respectively. 

Importantly, the acid- and reduction-induced instability could be further supported by TEM images. As shown in Figure 3F, after incubated in pH 5.0 for 24 h, the morphology of NPs was irregular, size distribution increased, and obvious aggregation could be observed. Interestingly, polyhedron appeared in the TEM image after 24 h incubation reductive condition (Figure 3G). As expected, size, size distribution, and morphology of the oDex-g-Pt+DOX NPs coexisted in pH = 5.0 with 10 mM GSH were sharply changed, which was ascribed to the quick release of dual drugs (DOX and Pt) (Figure 3H). Tyndall effect was further used to investigate the disassembly of oDex-g-Pt+DOX. As shown in Appendix A, oDex-g-Pt+DOX NPs had obvious tyndall effect. However, after treated in pH 5.0 with 10 mM GSH for 24 h, tyndall effect decreased sharply. The above results indicated the dissociations of oDex-g-Pt+DOX NPs in the acid and reductive environment.

### 3.4. In Vitro Drugs Release

DOX and Pt were conjugated to oDex via stimuli-responsive linkages and drug release behaviors of the oDex-g-Pt+DOX NPs were pH and reduction dual-sensitive. The in vitro drug release was detected at pH 5.0 or 7.4 to mimic acidic microenvironment of tumor intracellular components and normal pH of physiological conditions, respectively. Moreover, different pH with 10 mM GSH was used to mimic the reductive microenvironment of tumor intracellular components. As shown in Figure 4, no significant burst release of DOX and Pt was observed from oDex-g-Pt+DOX NPs. The cumulative DOX release from oDex-g-Pt+DOX NPs increased apparently as the pH decreased from 7.4 to 5.0 (Figure 4A). At physiological pH, there was only approximately 18.9% of DOX releasing from oDex-g-Pt+DOX NPs after 48 h. However, 51.4% of DOX was released at pH 5.0. The accelerated DOX release at pH 5.0 was attributed to the acid-cleavable oxime bond in oDex-g-Pt+DOX NPs. Incubation of the oDex-g-Pt+DOX NPs at pH 7.4 with 10 mM GSH lead to a slightly increased cumulative release of DOX (25.5%) after 48 h, and more than 79% of DOX was released at pH 5.0 with 10 mM GSH within the same period. This might be ascribable to the release of Pt, which induced the hydrophilic conversion processes of the NPs. During the Pt release procession, oDex-g-Pt+DOX NPs released more Pt as the addition of 10 mM GSH and decrease of pH. As depicted in Figure 4B, the release proportions of Pt from oDex-g-Pt+DOX NPs was faster and reached 56.4% in pH 7.4 with 10 mM GSH in 48 h, which was much higher than those in pH 7.4 (21.9%), and final release of Pt anticancer drug was approximately 77.3% after incubation with 20 mM GSH at the same time point (Appendix A), further indicating the reduction-responsive degradation of oDex-g-Pt+DOX NPs. As expected, the cumulative Pt release profiles indicated that the highest contents of released Pt from oDex-g-Pt+DOX NPs was 76.6% at pH 5.5 with 10 mM GSH after 48 h. The acidity-accelerated Pt release profiles might be owing to the pH-responsive DOX release and a more hydrophilic shell of NPs. These results proved that oDex-g-Pt+DOX NPs were comparatively stable under normal physiological conditions, while the chemical conjugated DOX and Pt were released by breaking the pH-sensitive hydrozone bond and reduction-sensitive Pt (IV) group, respectively. The pH and reduction dual sensitivity will endow the oDex-g-Pt+DOX NPs with high-selective intracellular release of DOX and Pt for combination chemotherapy.

### 3.5. In Vitro Cytotoxicity and Cellular Uptake

The promising and desirable result of a nDDS for chemotherapy is the maintenance (or reinforcement) of the drugs’ efficacy and, at the same time, the safety of carrier on healthy cells. Initially, the biocompatibility of Dex and oDex were investigated via MTT assays in L929 cells and HeLa cells. The results revealed that both Dex and oDex displayed no obvious cytotoxicity toward L929 cells and HeLa cells even at a high concentration of 1.0 mg/mL (Figure 5). These results indicated that both Dex and oDex had good biosafety.

The in vitro cell proliferation inhibition activity of various drugs and their nanoformulations were further evaluated against HeLa and A549 cells by MTT assay. The results are shown in Figure 6 and Appendix A. Obviously, with the increase of drug concentration and the evolution of time, the enhanced anti-proliferative activities were displayed toward all the test formulations. For instance, as shown in Figure 6A,B and Appendix A, after 48 h or 72 h co-incubation with HeLa cells, cytotoxicity order of various DOX and Pt formulations at dosages of 13.5 μM was as follows: Cis+DOX > Pt (IV)+DOX ≈ DOX > Cis > oDex-g-Pt+DOX > Pt (IV) > oDex-g-Pt. The free drugs (Cis, DOX and Cis+DOX) showed better in vitro anti-cancer efficacy than nanoformulations. This might be caused by the rapid passive diffusion of free drugs and the prolonged drug release from nanoformulations via endocytosis process. Cis+DOX indicated the highest cellular cytotoxicity, owing to the combinational effect between DOX and Pt. More fascinatingly, oDex-g-Pt+DOX NPs showed better inhibition efficacy compared with oDex-g-Pt NPs. In addition, the IC_50_ values of all drugs and nanoformulations decreased along with the extension of co-incubation time from 48 to 72 h. The oDex-g-Pt+DOX NPs displayed enhanced cellular cytotoxicity with IC_50_ values of 14.5 μM for 48 h and <3.375 μM for 72 h. Then, the in vitro cellular cytotoxicity of drugs and nanoformulations toward A549 cells were also detected, and similar results were obtained (Figure 6C and Appendix A). The findings demonstrated that the acid and reduction dual-sensitive oDex-g-Pt+DOX prodrugs showed great potential for combination chemotherapy.

Cisplatin is a common chemotherapeutic drug in clinic. However, its direct administration via intravenous injection causes serious side effects and severe drug resistance [36]. Previous reports demonstrated that synergetic delivery with DOX and cisplatin could work in concerted way or overcome drug resistance [34]. The cellular cytotoxicity of drugs and nanoformulations was further assayed using cisplatin-resistant cancer cells (A549/DDP) (Figure 6D and Appendix A). Notably, the IC_50_ value of cisplatin against A549 cells was 7.04 μM (Appendix A). In Figure 6D and Appendix A, A549/DDP cells showed obvious resistance to cisplatin and Pt (IV), the IC50 value of cisplatin and Pt (IV) against A549/DDP cells was both higher than 216 μM. As expected, DOX itself was more susceptive than cisplatin in A549/DDP cells, when cisplatin was used with DOX, the cell death was dramatically increased with significantly reduced IC50 values (15.5 μM). Meanwhile, Pt (IV)+DOX groups displayed high cytotoxicity towards A549/DDP cells. It was exciting to find that oDex-g-Pt+DOX NPs exhibited a closer in vitro cellular cytotoxicity to the Cis+DOX group. These results suggested that DOX and cisplatin had synergistic therapeutic effect, and oDex-g-Pt+DOX NPs might overcome the acquired drug resistance to cisplatin to some extent.

Subsequently, the in vitro cellular internalization and intracellular drug release behaviors of prodrugs drugs release performance of oDex-g-Pt+DOX NPs were examined in HeLa cells by CLSM. oDex-g-Pt+DOX NPs were incubated with HeLa cells for 0.5 h and 4 h, respectively. Hoechst 33,258 and red lyso-tracker were applied to stain the nucleus (blue imaging) and the lysosomes (green imaging), respectively. The subcellular localization of DOX was observed by the red fluorescence imaging. As shown in Figure 7, a time-dependent cellular internalization was clearly observed for oDex-g-Pt+DOX NPs, much higher DOX fluorescence intensities were exhibited at 4 h than at 0.5 h. A much weaker DOX fluorescence was detected after co-incubation with oDex-g-Pt+DOX NPs prodrugs for 0.5 h, and the DOX fluorescence was mainly co-localized within the lysosomes/endosomes in the cytoplasm. As the culture time was prolonged to 4 h, fluorescence signal of DOX in HeLa cells was significantly intensified, indicating the enhanced cellular uptake efficiency. Moreover, fluorescence of DOX in both cytoplasm and nuclei could be observed, and significant fluorescence of DOX was displayed at the perinuclear region, which indicated that oDex-g-Pt+DOX NPs were continuously transferred into the cells, and DOX was gradually released. The results proved that oDex-g-Pt+DOX NPs entered HeLa cells slowly via the endocytosis pathway, and DOX was sustainably released from the nanoformations.

The cell uptake of free DOX and oDex-g-Pt+DOX NPs toward HeLa cells for both 0.5 and 4 h was further analyzed by FCM. The fluorescent intensities of DOX were in the following order: oDex-g-Pt+DOX NPs group with 0.5 h incubation < DOX group with 0.5 h incubation < DOX group with 4 h incubation < oDex-g-Pt+DOX NPs group with 4 h incubation, as shown in Figure 8. After 0.5 h co-incubation, free DOX exhibited the stronger fluorescent intensity compared with oDex-g-Pt+DOX NPs, owing to quick diffusion pathway of free DOX. However, with the extension of culture time from 0.5 to 4 h, the oDex-g-Pt+DOX NPs exhibited upregulated DOX fluorescent intensity, which might be due to the intracellular drug accumulation from the sustained drug release of oDex-g-Pt+DOX NPs. The results indicated that the smart oDex-g-Pt+DOX NPs conjugates showed efficient endocytosis and intracellular drugs release.

## 4. Conclusions

In this work, an amphiphilic oDex-g-Pt+DOX prodrug was designed and synthesized as a smart drug delivery platform for combination chemotherapy application. oDex-g-Pt+DOX was developed by the versatile esterification reaction between the hydroxyl group of oDex and carboxyl group in Pt (IV) prodrug, and then the prepared polymer was conveniently functioned via the facile schiff base reaction between aldehyde groups of oDex and amino group in DOX. The obtained oDex-g-Pt+DOX conjugates spontaneously formed into NPs in aqueous medium with an average diameter of 183 nm. The smart oDex-g-Pt+DOX contained several fascinating advantages: (i) facile synthesis of the dual drugs delivery system with an esterification and schiff base reaction; (ii) plenty stability in the normal physiological environment without initial burst release; (iii) acid- and reduction-triggered dissociation, DOX and Pt release; (iv) besides, the multifunctional oDex-g-Pt+DOX NPs with its important merits including enhanced and combinational anti-tumor efficacy in vitro, especially in the cisplatin-resistant cell line. Therefore, the exploited acid and reduction dual-responsive oDex-g-Pt+DOX NPs exhibited great potential for clinical combination chemotherapy.

## Data Availability

Data is contained within the article or Appendix A.

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
