# Peer review of "pH and Reduction Dual-Responsive Bi-Drugs Conjugated Dextran Assemblies for Combination Chemotherapy and In Vitro Evaluation"

_polymers, 2021, doi:10.3390/polym13091515_

Round 1
Reviewer 1 Report
The paper submitted by Xue et al. deals with the synthesis and characterization of polymer/drug conjugates based on dextran and two drugs, Dox and Pt (IV). Even if the polymer/drug conjugates are well known in the literature, this study is original as it proposes the conjugation of two drugs on the same polymer backbone.
The manuscript is clear, well written, and the conclusions are supported by the results. However, some minor modifications are needed in order to increase the overall quality:
- The introduction section must be completed with several references concerning the methods of modification of Dex in order to obtain graft copolymers but also the use of Cisplatin as loaded drug. Some recent review articles might be useful: https://doi.org/10.3390/polym13030477; https://doi.org/10.1016/j.progpolymsci.2017.06.001; https://doi.org/10.3390/polym12051018
- Line 277: what is DHP?!
- In the literature there are some studies were the concentration of GSH was increased to 20 mM. Have the authors tried to use this concentration in order to increase the drug release efficiency?!
Author Response
Point 1: The introduction section must be completed with several references concerning the methods of modification of Dex in order to obtain graft copolymers but also the use of Cisplatin as loaded drug. Some recent review articles might be useful: https://doi.org/10.3390/polym13030477; https://doi.org/10.1016/j.progpolymsci.2017.06.001; https://doi.org/10.3390/polym12051018
Response 1: Special thanks to you for your good comment.
We have carefully studied the two reviews (https://doi.org/10.3390/polym13030477; https: //doi.org/10.1016/j.progpolymsci.2017.06.001) and the article (https://doi.org/10.3390/polym 12051018). Nanosized drug delivery systems based on natural biopolymers, particularly polysaccharides, were comprehensively highlighted in the two reviews. Moreover, cisplatin loaded polymeric nanoparticles was obtained by non-aqueous emulsion polymerization method. These reports are useful and helpful for the revision of our paper, and can provide a new research idea and method for us.
As your suggestion, three references have been cited in the revised manuscript as reference 42, 43, 46.
Point 2: Line 277: what is DHP?
Response 2: Thanks for your kindly reminder, it is very valuable for improving our paper.
DHP is the abbreviation of diamminedichloro-dihydroxyplatinum. We have expanded this abbreviation in the revised manuscript (Line 149).
Point 3: In the literature there are some studies were the concentration of GSH was increased to 20 mM. Have the authors tried to use this concentration in order to increase the drug release efficiency?!
Response 3: Great thanks to you for your good question.
It’s well known that the high concentration gradient of GSH between the intracellular (∼10 mM) and extracellular environment (∼2 μM) can be used as an ideal trigger to design redox responsive polymer prodrugs or micelles (https://pubs.acs.org/doi/abs/10.1021/acsami. 8b16363; https://dx.doi.org/10.1021/acs.biomac.9b01578). 10 mM GSH was usually used to mimic the intracellular microenvironment of tumor. Certainly, in some studies, the concentration of GSH was increased to 20 mM (https://pubs.acs.org/doi/10.1021/acs.biomac.7b00445). According to the literature, compared to 10 mM GSH, the drug released from reduction-sensitive nanocarrier was rapidly accelerated in the presence of 20 mM GSH.
According to your suggestion, 20 mM GSH was further used to evaluate the reduction stimuli-responsive Pt release behaviors. The results were shown in Figure R1 (Figure S8 in SI), the experiment and data were added in the corresponding sections. The revisions have been highlighted in red color.
Figure R1. Drugs release profiles. Pt release behaviors from oDex-g-Pt+DOX NPs in the presence of 10 mM or 20 mM GSH.
As shown in Figure R1, final release of Pt anticancer drug from oDex-g-Pt+DOX NPs was approximately 56.4% after incubation for 48 h in the presence of 10 mM GSH, but was about 77.3% in the presence of 20 mM GSH at the same time point. The accelerated Pt release under 20 mM GSH likely reflects the rapidly reductive cleavage of Pt (IV).
According to your comments, we have revised the paper extensively. Thanks again for your help.

Reviewer 2 Report
Paper titled (Acid and Reduction Dual-sensitive Dextran-di-drugs Conju- 2
gate Assemblies for in vitro Combination Chemotherapy) by Xue et al., studied the synthesis of new formula of chemotherapy combination (doxorubicin & cisplatin) and tested its efficacy.
Title: needs revision
Abstract: (However, the conventional polymeric prodrug with a single drug still suffers from unsatisfied efficiency owing to the heterogeneity of cancer and occurrence of drug resistance): I do not think this statement is correct.
Introduction: it seems non of the authors is a biologist or oncologist. The information in the first paragraph is not totally true! needs replacement by correct info.
Authors used some abbreviations such as NPs, then did not utilize them later. Kindly revise the use of all abbreviations.
Some paragraphs are too long and needs to be dividied into 2.
The Intro is too long in general and needs to be shortened to be more concrete.
3- Methods: give the rational for the selection of the 2 pH values at which the PDI was calculated.
4- There is a missed control group for oDex-g-DOX
5-The major drawback in this article that there is nothing about statistical analysis between groups in pharmaceutical parameters nor in vitro study. Hence one cannot judge the data or the preference of one formula over another. Unfourtnately, this limits the usefulness of this study.
Author Response
Response to Reviewer 2 Comments
Point 1: Title: needs revision
Response 1: Great thanks to you for the suggestion about the title.
After careful consideration, the title of paper has been revised as “pH and Reduction Dual-responsive di-drugs Conjugated Dextran Assemblies for in vitro Combination Chemotherapy”.
Point 2: Abstract: (However, the conventional polymeric prodrug with a single drug still suffers from unsatisfied efficiency owing to the heterogeneity of cancer and occurrence of drug resistance): I do not think this statement is correct.
Response 2: Special thanks to you for your good comment.
This statement in “abstract” section has been revised as follows: However, the conventional polymeric prodrug with a single drug still suffers from unsatisfied efficiency.
Point 3: Introduction: it seems non of the authors is a biologist or oncologist. The information in the first paragraph is not totally true! needs replacement by correct info.
Response 3: Thank you very much for your comments on our manuscript.
We have read the relevant articles, and carefully revised the first paragraph in “introduction” section. The revisions have been highlighted in red color.
Point 4: Authors used some abbreviations such as NPs, then did not utilize them later. Kindly revise the use of all abbreviations.
Response 4: Great thanks to you for your kind suggestion, it’s helpful for the revision of our paper.
NPs is the abbreviations of nanoparticles, the nanoparticles appeared in the paper later was revised to NPs. DOX is the abbreviations of doxorubicin, and we also utilized the abbreviated form in the paper. All the revisions have been highlighted in red color in the manuscript.
Point 5: Some paragraphs are too long and needs to be dividied into 2.
Response 5: Thank you very much for your kind reminder.
The “Stability of oDex-g-Pt+DOX NPs” in the “results and discussion” section is too long, this paragraph has been dividied into 3 short paragraphs.
Point 6: The Intro is too long in general and needs to be shortened to be more concrete.
Response 6: Special thanks to you for your good comment and suggestion.
As suggestions, we have revised the introduction according to your suggestion. All the revisions have been highlighted in red color.
Point 7: Methods: give the rational for the selection of the 2 pH values at which the PDI was calculated.
Response 7: Great thanks to you for your question.
In our paper, DOX was conjugated onto the oDex-g-Pt via Schiff base reaction, the formed hydrazone linkages are acidity-labile. Then, two pH values, pH 5.0 and pH 7.4, were selected for the stability evaluation of the prepared oDex-g-Pt+DOX NPs. pH 5.0 and pH 7.4 were used to mimic the tumor intracellular components and normal physiological conditions, respectively. The two pH values are usually used to study the pH-sensitive property of nanodrug delivery system (https://pubs.acs.org/doi/abs/10.1021/acs.biomac.5b00625; http://pubs.acs.org/doi/10.1021/acsami.8b16363).
Point 8: There is a missed control group for oDex-g-DOX.
Response 8: Great thanks to you for your comment, it’s very valuable for revising our paper.
Dextran and its derivatives have been extensively investigated as nanocarriers for cancer imaging and therapy. DOX conjugated onto the oxidized dextran (oDex) with abundant aldehyde groups had been successfully synthesized and reported by other works. Chen’s group had designed and synthesized two intracellular acid-sensitive dextran−doxorubicin (Dex−DOX) conjugates through the Schiff base reaction between the aldehyde groups in the oDex with different lengths and the amino group of DOX (Figure R1) (http://dx.doi.org/10.1016/j.carbpol.2016.12.070). The self-assembly properties, various intracellular acid-accelerated release behaviors, and even adjustable antitumor efficacies in vitro and in vivo were comprehensively studied. Moreover, a novel self-assembling prodrug containing pH- and redox-responsive functional groups was prepared by covalent conjugation of DOX and lipoic acid to the oDex (https://doi.org/10.1016/j.colsurfb.2019.110537) (Figure R2).
Figure R1. Preparation and metabolism in vivo. Schematic illustration for self-assembly, in vivo circulation, accumulation in tumor tissue, and final pH-triggered intracellular DOX release after intravenous injection of pH-responsive oDex−DOX prodrug.
Figure R2. (B) Esterification of oDex with lipoic acid, (C) DOX attachment by Shiff base formation.
According to your suggestion, we prepared the control group oDex-g-DOX (Figure R3) and oDex-g-DOX NPs. oDex-g-DOX was characterized by 1H NMR (Figure R4) and FT-IR spectra (Figure R5). The UV-vis absorbance (Figure R6), size and size distribution (Figure R7), and in vitro cytotoxicity (Figure R8) were carefully investigated. The results were as follows:
Figure R3. Synthetic scheme of oDex-g-DOX.
This figure was added as Scheme 2 in the “Supporting Information”.
Figure R4. The 1H NMR spectrum of oDex-g-DOX in DMSO-d6.
This figure was added as Figure S1C in the “Supporting Information”.
Figure R5. Fourier-transform infrared spectra (FT-IR) spectrum of the oDex-g-DOX.
The data were added into Figure S3 in the “Supporting Information”.
Figure R6. UV–visible spectrum of oDex-g-DOX NPs in aqueous solution (A) and DMSO (B).
The data were added into Figure S4 in the “Supporting Information”.
Figure R7. Size and size distribution of prepared oDex-g-DOX NPs.
This figure was added as Figure S5 in the “Supporting Information”.
Figure R8. Cell viability curves of HeLa cells incubated with oDex-g-DOX.
The data were added into Figure S9A and 9B in the “Supporting Information”.
All the corresponding revisions were highlighted in red color in the “Manuscript” and “Supporting Information”.
Point 9: The major drawback in this article that there is nothing about statistical analysis between groups in pharmaceutical parameters nor in vitro study. Hence one cannot judge the data or the preference of one formula over another. Unfourtnately, this limits the usefulness of this study.
Response 9: Thank you for your careful inspection.
According to your suggestions, we redrawn Figure 6 (Figure R8), and made statistical analysis. We also added the corresponding informations in the manuscript, and the revisions were highlighted in red color.
Figure R8. In vitro anti-cancer efficacy of drugs and nanodrugs. Cell viability curves of HeLa cells incubated with drugs and nanodrugs for 48 h (A) or 72 h (B). Cell viability curves of A549 cells (C) and A549/DDP cells (D) after incubation with drugs and nanodrugs for 48 h. Statistical significance was calculated via one-way ANOVA analysis with a Tukey post-hoc test. *p < 0.5, **p < 0.01, ***p < 0.001, ****p < 0.0001, ns: not significant.
Thanks again for your good comments and kind suggestions, which are very helpful for the revision of our paper.

Round 2
Reviewer 2 Report
Although authors improved the paper partly -thanks- and replied to some questions correctly, the answers did not appear in the paper itself.
1- For example : the previous question on rationalizing the pH selection
2- Statistical analysis was not satisfactorely done to all possible parameters
3- Data in Figure 6 : no need to mention what is ns : just highlight the significant differences.
4- student's t test is not the correct test for analyzing these data.
ANOVA followed by post hoc test are the suitable tests.
5- In all illustrations, they should be self explanatory, please mention the full name for each compound
6-In all illustrations, mention the type of data & stat test used for analysis and sympols.
7- For abstract and INTRO, (previous comment also) : there is no disadvantage for the monodrug nanoparyicles. BUT the technique can be used for combined drugs such as your current work. Please consider this overall the manuscript
8- Title: I think bidrug will be better than didrug
I suggest the title to be:
pH and Reduction Dual-responsive bi-drugs Conjugated Dextran Assemblies for Combination Chemotherapy and in vitro evaluation.
9- Figure 4: the x axis unit is hours? please confirm
10-Figure 4: please statistically analyze at least the last point at 50
11-Please put the synthetized molecules in a table with clear nomenclature for each moelcule
12- Why a Cis group suddenly appeared in the in vitro assay? what is its role and how synthetized?, but cannot find it in methods? which group is this?
A general remark: it is not fine to say thank you for the reviewer on each question reply, somebody may find it an exaggerated response
Please organize the paper well to be easlily comprehended by the reader
Author Response
Response to Reviewer 2 Comments
Point 1: For example: the previous question on rationalizing the pH selection.
Response 1: The rational for the selection of the 2 pH values (pH 7.4 and pH 5.0) was added in the revised manuscript (Line 338-342 in the manuscript).
Point 2: Statistical analysis was not satisfactorely done to all possible parameters.
Response 2: According to your suggestion, we have made statistical analysis for Figure 4A and 4B. The results were displayed as follows:
Figure 4. Drugs release profiles. (A) DOX and (B) Pt release behaviors from oDex-g-Pt+DOX NPs (platinum plus DOX conjugated oxidized dextran nanoparticles) at different conditions. Data points represent mean ± s.d. (n = 3) from three independent experiments. Statistical significance was calculated via one-way ANOVA analysis with a Tukey post-hoc test. *p < 0.5, **p < 0.01, ***p < 0.001, ****p < 0.0001.
We also have revised the manuscript, and the revisions were highlighted in red color.
Point 3: Data in Figure 6: no need to mention what is ns: just highlight the significant differences.
Response 3: All the “ns” in Figure 6 were deleted.
Figure 6. In vitro anti-cancer efficacy of drugs and nanodrugs. Cell viability curves of HeLa cells incubated with drugs and nanodrugs for 48 h (A) or 72 h (B). Cell viability curves of A549 cells (C) and A549/DDP cells (D) after incubation with drugs and nanodrugs for 48 h. Data points represent mean ± s.d. (n = 3) from three independent experiments. Statistical significance was calculated via one-way ANOVA analysis with a Tukey post-hoc test. *p < 0.5, **p < 0.01, ***p < 0.001, ****p < 0.0001.
Point 4: student's t test is not the correct test for analyzing these data.
ANOVA followed by post hoc test are the suitable tests.
Response 4: We have revised the “Statistical analysis” section as follows:
All the experiments had three replicates (n = 3) at least. Data were presented as mean standard deviation (SD). The one-way ANOVA analysis with a Tukey post-hoc test was carried out to analyze the statistical significance: *p < 0.05, **p < 0.01, ***p < 0.001, ****p < 0.0001.
Point 5: In all illustrations, they should be self explanatory, please mention the full name for each compound.
Response 5: In all illustations, we have modified the full name for each compound. All the revisions were highlighted in red color.
Point 6: In all illustrations, mention the type of data & stat test used for analysis and sympols.
Response 6: We have revised the illustrations in Figure 4 and Figure 6 according to your suggestions.
Point 7: For abstract and INTRO, (previous comment also): there is no disadvantage for the monodrug nanoparyicles. BUT the technique can be used for combined drugs such as your current work. Please consider this overall the manuscript
Response 7: As you say, there is no disadvantage for the monodrug nanoparticles, but the technique can be used for combined drugs. Therefore, we deleted the inappropriate expressions.
Point 8: Title: I think bidrug will be better than didrug
I suggest the title to be: pH and Reduction Dual-responsive bi-drugs Conjugated Dextran Assemblies for Combination Chemotherapy and in vitro evaluation.
Response 8: After consideration, the title was revised to “pH and Reduction Dual-responsive bi-drugs Conjugated Dextran Assemblies for Combination Chemotherapy and in vitro evaluation”.
Point 9: Figure 4: the x axis unit is hours? please confirm.
Response 9: We confirmed that the x axis unit in Figure 4 is hours.
Point 10: Figure 4: please statistically analyze at least the last point at 50
Response 10: The statistical analysis has been made for Figure 4.
Point 11: Please put the synthetized molecules in a table with clear nomenclature for each moelcule
Response 11: Thank you very much for your suggestion. We synthesized two platinum derivatives, diamminedichloro-dihydroxyplatinum (c,c,t-[Pt(NH3)2Cl2(OH)2], DHP) (Line 150 in the manuscript) and succinic anhydride modified DHP (c,c,t-[Pt(NH3)2Cl2(OH)(O2CCH2CH2CH2CO2H)], Pt (IV) prodrug) (Line 151 in the manuscript). The full name and its corresponding abbreviation were listed in Table 1. Table 1 was added into the manuscript.
Table 1. The full name and corresponding abbreviation for the synthesized molecules and polymers.
|
Full name |
Abbreviation |
|
diamminedichloro-dihydroxyplatinum |
DHP |
|
succinic anhydride modified DHP |
Pt (IV) |
|
oxidized dextran |
oDex |
|
platinum conjugated oxidized dextran |
oDex-g-Pt |
|
DOX conjugated oxidized dextran |
oDex-g-DOX |
|
platinum plus DOX conjugated oxidized dextran |
oDex-g-Pt+DOX |
Point 12: Why a Cis group suddenly appeared in the in vitro assay? what is its role and how synthetized? but cannot find it in methods? which group is this?
Response 12: Cis is the abbreviation of cisplatin (Line 223 in the manuscript). Cisplatin is the raw material, which is used to synthesize the platinum derivative, DHP and Pt (IV). Cis group was appeared in the in vitro assay in Figure 6D to evaluate the cisplatin resistance in cisplatin-resistant lung carcinoma A549 cells (A549/DDP cells). For other cisplatin sensitive cell lines, HeLa and A549 cells, the in vitro cytotoxicity assay of cisplatin was also investigated (Figure S9A, 9B and 9C).
Thank you very much for your suggestions and comments.

Round 3
Reviewer 2 Report
I recommend acceptance of the current form